# An Adapted Model for Transition to Adult Care in Young Adults with Prader–Willi Syndrome

**DOI:** 10.3390/jcm10091991

**Published:** 2021-05-06

**Authors:** Maria Pedersen, Charlotte Höybye

**Affiliations:** 1Department of Endocrinology, Karolinska University Hospital, 171 76 Stockholm, Sweden; maria.pedersen@sll.se; 2Department of Molecular Medicine and Surgery, Karolinska Institute, 171 76 Stockholm, Sweden

**Keywords:** Prader–Willi syndrome, transition to adult care, endocrinology

## Abstract

Background: Prader–Willi syndrome (PWS) is a rare, neurodevelopmental, genetic disease caused by the lack of expression of paternal genes in chromosome 15. The typical characteristics, including hyperphagia, muscular hypotonia, abnormal body composition, hormonal deficiencies, cognitive disabilities, and behavioral problems, appear or worsen in young adults, and the development of comorbidities increases. The transition of care of young adults with PWS is a challenge due to the complexity of the disease and the vulnerability of the patients. Multidisciplinary transition clinics are optimal but difficult to implement in clinics with few transitions. Methods: The description of transition care meetings was limited to the pediatric and adult endocrinologists and nurses. The presentation of our checklist was developed to optimize the organization of the transition of young adults with PWS. Results: Two to four patients with PWS are transferred to adult care every year in our hospital. Transition with the adapted program was more comfortable for patients and identification of the individual patient’s comorbidities and special needs could clearly be addressed. Conclusions: In smaller settings, an adapted model including a proper introduction and presentation of the adult team and clinic, careful information about comorbidities and special needs, together with a checklist can optimize the transition of care to adult care.

## 1. Introduction

Prader–Willi syndrome (PWS) is a rare, neurodevelopmental, genetic disease caused by the lack of expression of paternal genes in the PWS region of chromosome 15 (15q11.2-q13) [1]. In 65–70% of patients, the syndrome is caused by a deletion of paternal genes in the region; in 20–30%, by a maternal uniparental disomy (UPD); and in the remaining 2–5%, by an imprinting defect [1]. However, in a recent study, a change was found in the distribution of the genetic causes, with a larger proportion of patients found to have UPD [2]. In adolescents and young adults, the syndrome is typically characterized by hyperphagia, muscular hypotonia, abnormal body composition with more body fat than lean body mass, hormonal deficiencies, cognitive impairments, and behavioral problems [3,4]. Many of the characteristics appear or worsen during adolescence and early adulthood, and the development of comorbidities such as diabetes, hypertension, and sleep apnea increases [4]. Hypogonadism and growth hormone (GH) deficiency are the most common hormone deficiencies. 

Transition is commonly defined as a set of physical and psychological changes starting in late puberty and ending with full adult maturation [5]. The start of transition is usually in the mid- to late teens, and the end of transition comes 6–7 years after the achievement of final height [5]. This period includes many somatic, psychological, and lifestyle changes and is often a challenging period, especially for patients with chronic diseases and disabilities. For many years, it has been well known that young adults with chronic diseases need dedicated health care programs when care is transferred from the pediatric to the adult clinic to facilitate optimal continued care [5]. Ideally, the process should start in early adolescence, enabling the young patient to establish a relationship with the adult team prior to the day of the transfer. In many countries, specialized, multidisciplinary clinics have been established, for example, the “Ready, Steady, Go” program from Southampton, UK, which is used as a transition model in several centers [6].

PWS is a complex and multisystemic disease, and the optimal care involves a multidisciplinary team of several specialists [3,4]. PWS is usually managed by the same pediatric team from birth to the time for transition to adult care. During childhood, the multidisciplinary team is often well established and has been well known to the patient for many years. Adult care is usually not as integrated and often consists of different specialists working in different departments or even in different hospitals. For all ages, the focus of care is on the prevention and treatment of comorbidities, as well as on auxology during childhood [7]. Furthermore, in the adult care setting, patients are expected to have sufficient maturity to take more responsibility for the implementation of their own care than they are in the pediatric care setting. Because of the complex medical conditions and psychological and behavioral problems in PWS, the continuation of the care implemented by the pediatricians is important. Therefore, several considerations as well as flexibility are required for the transition of care of patients with PWS.

In the literature, collaboration between pediatric and adult multidisciplinary teams is recommended for an optimal transition of young adults with PWS [3,4,7]. It has clearly been shown that a coordinated care pathway with specialized pediatric care and an organized transition from a pediatric hospital to a reference center for PWS is associated with improved endocrine, metabolic, anthropometric, and psychiatric traits in adulthood [3]. In many hospital settings, integrated, multidisciplinary care by a well-established team cannot be continued in adulthood for several reasons, and a different organization of the transition is required. Here, we describe our experience with the organization of transition care meetings in a limited setting with pediatric and adult endocrinologists and nurses. Furthermore, we present our checklist developed for optimizing the organization of the transition of young adults with PWS.

## 2. Development of an Adapted Model for Transition and a Checklist

Our hospital is a tertiary reference hospital for highly specialized care. Currently, 70 children with PWS are seen by a pediatric specialist and 42 by an adult endocrinologist, both with long experience of PWS. During the past 10 years, two to four young adults have been transferred from pediatric to adult care per year.

The age at which the patients with PWS have been transferred from the pediatric to the adult endocrinology clinic has decreased. For the past 10 years, the youths have been transferred at the same age as youths with other chronic diseases, which in our hospital is at 18 years of age. Thus, most of the patients still attend school and live with their families, whereas previously, most had already moved from their parental home and had a daily occupation. As it is a multisystemic disease, many young adults with PWS are cared for by several specialists and some are treated with medications that must be monitored and not discontinued. Most of the young adults are not able to give or receive information about their diseases and treatments, and they cannot be expected to be fully involved in future planning. Therefore, transitions in the past became easily disorganized, and important aspects of the transition were not noticed until later. Furthermore, the different organizations of visits and examinations in the two settings also presented challenges for the transition process.

The pediatric and adult PWS team in our hospital consists of a nurse and an endocrinologist, both specialized in PWS for many years. Due to the complexity of PWS, multiple specialists collaborate on the patients’ care. In pediatric care, the different specialists, nutritionists, and physiotherapists are part of one large department, while adult care is split up in different departments and locations. The number of transitions per year of young adults with PWS is limited, and for many years, the transition consisted of the adult endocrinologist attending the patient’s last visit at the pediatric department in order to meet the patient and family as well as to learn important information. Tight working schedules for professionals and parents resulted in difficulties in finding a time for a transition meeting where all specialists were available and present, and a transition meeting was sometimes delayed or, for a few patients, did not occur. Although the medical records were carefully reviewed, important information, especially on nonmedical issues, was not completely conveyed. In addition, the young adults were not sufficiently prepared for the first visit at the adult clinic, the location was new, they had not met the nurse or other staff, and they lacked knowledge of new routines, issues that often led to anxiety and stress. New examinations were introduced, different to the ones they were familiar with, and at the same time, other routines they were familiar with were no longer performed. Altogether, many new things happened, routines for the visit changed, and patients were not always able to handle the situation.

## 3. Results

Based on these issues and experiences, we changed the organization of the transition visits two years ago. The adult endocrinologist and nurse now both participate in the last visit with the pediatrician, the patient, and the relatives. In our experience, it is very important that both the adult endocrinologist and nurse attend and be introduced at the meeting to increase the patient’s comfort. Important medical, psychological, and behavioral issues are discussed as well as future plans. The patient undergoes a physical examination performed by the pediatric endocrinologist. In cases where the patient has comorbidities such as diabetes, scoliosis, neurological or breathing problems, a more focused, relevant examination is performed. The patient’s medication is reviewed, and doses discussed and evaluated. In general, young PWS patients are healthy, but sometimes, a referral to another adult specialist is needed.

The main role of the adult endocrine nurse in the transition meetings is to establish a relationship with the patients and families as the nurse will be the first-line contact in the adult clinic. The goal is to provide a safe environment at the adult clinic with the patient always knowing whom to contact and meet, thus reducing the risks of unforeseen events that might distress the patients. We have found it beneficial to limit nurse contacts to one specific nurse for all patients with PWS to better provide continuity and distinctness. This might be particularly beneficial when comorbidity in the autism spectrum is present. In addition, the role of the nurse is to identify potential areas of care where information must be forwarded or the patient transferred to other settings, such as the district nurse. At the adult clinic, the nurse introduces the patient to the routine care performed there, such as bioimpedance, waist circumference, and blood pressure measurements. When a patient is treated with growth hormones, future nurse visits include follow-ups regarding injection sites, management, and eventual updates of current injection device.

In view of the complex nature of PWS, input from and cooperation among multiple specialists are necessary. Whenever required, other specialists, such as psychiatrists and/or psychologists or nutritionists, should be integrated in the transition meetings. After the visit, the patient and relatives visit the adult clinic, meet the staff, and receive information on the organization of care. Since the transition meetings are limited to joint meetings between the patient, relatives or caregivers, and the pediatric and adult teams, they are now performed without problems or delays and are never missed. In our hospital, the endocrinologist has the main responsibility for the care of the patient with PWS. Many young adults with PWS also need continuous care by other specialists, for example, a psychiatrist, a psychologist, a neurologist, an orthopedic surgeon, or a general practitioner. Moreover, continuous contact with nutritionists and physiotherapists is often required. In the Swedish healthcare system, these contacts for adults must be separately organized. This has sometimes already been arranged, but support is usually needed.

As PWS is a multisystemic disease, many issues exist, and it is easy to forget important things. Based on the “Ready, Steady, Go” program [6], we developed a short checklist covering the most important information of relevance for the care of the patient (Table 1). The checklist includes individual care and considerations and is filled in by the adult nurse at the last visit in the pediatric clinic. However, the checklist is appropriate for use by any practitioner. We have used this organization of transition meetings and the checklist for a few years. Eight patients have been transferred using the current model, and in our opinion, it works well. In our experience, the patients are now better prepared and less anxious. Interestingly, we have noticed that some of the patients searched the internet for information and photos of us and were happy when they recognized us at the transition visit. Furthermore, the future care by other specialists has become better organized and an initiative to plan for future living and occupation after school has been developed. We have never had dropouts from the visits in adult care, but with the current model, the first visit in the adult care setting is timelier and more patient centered.

## 4. Discussion

PWS is a multisystemic disease, and there are several health issues to consider in the transition of care from the pediatrician to the adult endocrinologist. The care initiated by the pediatrician needs to be continued in the adult care setting. It is important to understand the youth’s life situation and map health and risk factors, compliance to diet, and physical activity. Information on future care and discussion of expectations are also important. Transition between multidisciplinary teams is clearly preferred, but when this is not possible, a transition performed by a more limited team with the help of a checklist can still work well.

Young adults with PWS are confronted with several new requirements, challenges, and responsibilities. Neurocognitive dysfunctions such as poor concentration, endurance, memory, attention, and flexibility become increasingly evident and problematic, although most of the patients do not report the problems themselves [3,7]. In situations with increased stress and uncertainty, a tendency for temper outbursts, repetitive and ritualistic behaviors, and skin picking may occur more frequently. Psychosocial support is needed for the majority due to behavioral problems and sometimes psychiatric diseases. Young adults with UPD have a higher prevalence of psychoses and autism but better verbal skills, and the differences between patients with UPD and deletions become more apparent during adolescence and early adulthood [8]. Most of the young adults express a wish to live on their own and get a job, but in general, adults with PWS are unable to live independently and cannot manage competitive jobs [7].

The natural history of PWS includes six nutritional phases from birth to adulthood [9]. At the time of transition, the classical and complete phenotype with hyperphagia and poor satiety (phase 3) is present, and the risk of obesity is high if a controlled and restricted diet is not followed. Diet, physical activity, and weight are closely linked, and physical activity should be encouraged, sometimes with support of a physiotherapist. The ubiquitous risk of obesity in PWS increases the risk of development of diseases secondary to obesity, such as diabetes and cardiovascular diseases, and the weight prevention provided by physical activity is therefore important. Because of the increased risk of low bone mineral density and high frequency of scoliosis in PWS, physical activity is particularly important for these patients [3,7,10]. Furthermore, physical activity might have a positive impact on mental health, the ability to concentrate and learn [11], and the level of endorphins [12]. Comorbidities secondary to obesity should be managed according to general guidelines. Sleep-related breathing disorders are common in PWS, affecting falling asleep, sleep patterns, waking up in the morning, and wakefulness during daytime [4,10]. Obesity is a risk factor for sleep apnea and should also be prevented for this reason.

Since 2000, GH treatment in children with genetically confirmed PWS without prior GH stimulation testing has been approved in many countries [13]. GH promotes growth in children, develops bone strength, increases lean body mass, and reduces fat mass. In addition, it has beneficial effects on blood lipids and improves quality of life. Somatic and psychological development continues beyond adolescence and GH’s effects on body composition and metabolism continue in adult life [13]. However, GH treatment of patients with PWS after adult height is obtained is only approved in a few countries and reassessment of continued GH treatment is performed when the final height is reached [13]. In a recent placebo-controlled GH crossover trial, it was shown that GH-treated young adults with PWS who had attained adult height benefited from the continuation of GH treatment [14]. Thus, fat mass increased during placebo treatment, whereas fat mass decreased, and lean body mass increased during GH treatment [14]. In addition, studies in the transition phase have shown a beneficial effect of GH treatment on cognition [15] and metabolic profile [16]. Thus, GH treatment in childhood and adolescence has significantly changed the bodily appearance of children, adolescents, and young adults. Major safety concerns have not been noticed, but a reduction of the GH dose will be needed over time.

The treatment of other endocrine deficiencies, of which treatment with sex steroids is the most common, should also be addressed during transition. The majority of patients with PWS are hypogonadal. A combined form of hypogonadism is often present in PWS, but the degree of hypogonadism is variable and less severe in women [17,18]. Fertility has not been described in men with PWS, while some women with PWS are fertile and four pregnancies in PWS women have been documented worldwide [19]. There are no studies on treatment with sex hormones during transition, but in both genders, sex hormone replacement is important for the development of an adult appearance, muscle mass, and bone mineral density. An individualized approach to sex hormone treatment is necessary, and it is important to understand the distinction between treatment of hypogonadism and birth control management. Despite the high frequency of hypogonadism, many adults with PWS have strong romantic thoughts and an interest in sexual experiences. Fertility in PWS includes medical and ethical considerations. Appropriate anticipatory guidance, counseling, and education is important. Articles and reports on the four documented pregnancies in women with PWS indicate that the gestations were uncomplicated, and three babies were delivered by planned caesarean sections [20]. The mothers were unable to breastfeed and bond with the children, and the infants were mostly taken care of by others.

Various guidelines for the transitioning of the care of adolescents with chronic medical conditions to the care of adult specialists have been published [5,21]. Furthermore, previous publications have described specific details of the different protocols and effects of transition programs for PWS in various institutions [3,4,7,8]. For an optimal and successful implementation of transition care, three practical points have been suggested: (1) patient-centered support and monitoring, (2) hospital-centered infrastructures with key personnel, and (3) flexibility in planning and modifying health care transition procedures [22]. Even though we do not have a specialized transition clinic or a multidisciplinary team, in our experience, our transition model and checklist can manage these three points properly without a multidisciplinary team present at the transition meeting. As noted by others, there might also be financial problems with having personnel work only in transition care, especially in rare diseases with a limited number of transitions per year [23].

## 5. Conclusions

In conclusion, the transition to adult care of young adults with PWS is a challenge due to the complexity of the disease and the vulnerability of the patients. Specialized, multidisciplinary clinics are preferable, but in clinics with few transitions, adapted models are needed. The concept of transition care is applicable to every institution caring for individuals with PWS, but specific protocols need to be adapted and modified according to the available staff and facilities of each clinic setting. A proper introduction and presentation of the adult team and the adult clinic, careful information about any comorbidity and special needs, together with a checklist can optimize the transition of care in smaller settings.

## Figures and Tables

**Table 1 jcm-10-01991-t001:** Checklist for transition from pediatric to adult care of young adults with Prader–Willi syndrome.

Name:	Birth Date:	Date:
Contact information:		
Contact information for relatives:		
Current/future residential situation:		
Contact information of group home/other:		
District nurse/home care:		
	Needs help with:	Comments:
Health and medication		
Injections? Preparation and dose:		
Other regular medication and dose?		
Renewal of prescriptions?		Informed about routines
Healthcare follow-up visits booked?		Informed about routines
Contact with psychiatrist?		
Contact with nutritionist?		
Contact with other specialists?		
Lifestyle		
Training/everyday exercise:		
Sleep:		
Nicotine habits:		
Meal planning/situation:		
Alcohol habits:		
Activity of daily living		
Hygiene, dressing/undressing:		
Daily transportation:		Transportation service
Work/education:		
Leisure time:		
Transition to the adult clinic		
Impedance measurements:		Informed about routines
Visit to the adult clinic and meet the staff		Hand out contact information

## Data Availability

The data are not publicly available due to privacy and ethical restrictions. The data that support the findings of this study are available on request from the corresponding author.

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
