# Peer review of "An Adapted Model for Transition to Adult Care in Young Adults with Prader–Willi Syndrome"

_jcm, 2021, doi:10.3390/jcm10091991_

Round 1

Reviewer 1 Report

This is an good paper about transition period in PWS patients that I think that deserves to be published. Some minor issues to be considered:

-page 2: "the main focus on care during childhood is on auxology": I think this minimizes the multidiscipinary work that it's made with these patients (neurologist, genetician, pediatric endocrinologist, nutritionist, psycologist and so on) The sentenceshould be redefined

-page 3: dropout s: should be dropouts

Author Response

Rewiever 1:

This is a good paper about transition period in PWS patients that I think that deserves to be published. Some minor issues to be considered:

-page 2: "the main focus on care during childhood is on auxology": I think this minimizes the multidiscipinary work that it's made with these patients (neurologist, genetician, pediatric endocrinologist, nutritionist, psycologist and so on) The sentence should be redefined

Response: We thank the reviewer very much for the nice and positive comments. For the comment regarding page 2 the importance of multidisciplinary care is mentioned in the lines above and the sentence has been modified to: “In all ages focus of care is on prevention and treatment of co-morbidities and in addition on auxology during childhood

-page 3: dropout s: should be dropouts

Response: Our apologies for this error. It has been corrected.

Reviewer 2 Report

This is a beautiful  and interesting paper that could help both specialized endocrinologist in Prader-Willi syndrome and nurses. The authors described their model for transition and checklist.

The article is well structured and the discussion is a beautiful review of the natural history of the syndrome.However, I think there are a little mistake in the introduction:-“ genes in the PWS region of chromosome 15 (15q11-q12) “  should be:“genes in the PWS region of chromosome 15 (15q 11.2-q13) according to the reference 1 (Cassisy SB et al 2012) 

Author Response

Reviewer 2:

This is a beautiful and interesting paper that could help both specialized endocrinologist in Prader-Willi syndrome and nurses. The authors described their model for transition and checklist.

The article is well structured and the discussion is a beautiful review of the natural history of the syndrome.However, I think there are a little mistake in the introduction:-“ genes in the PWS region of chromosome 15 (15q11-q12) “  should be:“genes in the PWS region of chromosome 15 (15q 11.2-q13) according to the reference 1 (Cassisy SB et al 2012) 

Response: We thank the reviewer very much for the positive comments. The error has been corrected.

Reviewer 3 Report

Manuscript by Maria Persen et al describes the importance of a well-structured transition, from pediatric to adult care, in young adults with Prader-Willi syndrome. 

The authors state that, despite the importance of the collaboration between the pediatric and the adult team for an optimal transition, joined medical visits at the time of transition are extremely difficult given the tight schedule of the medical teams in a hospital setting. The fact of not being able to have a joined visit often results in miss information and stress and anxiety of the unprepared patients for the new situation.  To resolve the situation the authors present a new model of transition.

Concerns:

  • Please described the new model with details
  • Please give a full prospective on the transition and include the pediatric team prospective. What is the pediatric team doing at the last visit? Authors state that the main focus on the care during childhood is auxology. Please review with the pediatric team as the pediatric team main focus is mainly the prevention of co-morbidities and also treatment when needed and not auxology.
  • Please provide more details on important medical facts at the time of transition. Otherwise review the manuscript as role of the nurses in the transition of PW patients
  • The new model is based on a joined visit with the pediatric and adult team? How the authors resolved the possibility of performing a joined visit with their tight schedule? Are they still having the same trouble and often end up missing that joined visit?
  • They have adapted a check list from the “Ready, Steady, Go” program to not miss any information. Is the pediatric team filling the check list and then the nurse from the adult team reviewing?
  • Results: the results should be carefully reviewed. 1)How many patients have you transitioned using the current model?2)Authors state that patients are now prepared and less anxious; how have the authors measured this outcome? 3) Authors state they have no dropouts with the current model; did you have any dropouts before? Please provide more information about the experience
  • Please review title: “an adapted model”? is it an adapted model or an adapted check list?

Authors need to provide all the readers with the following information: details about the transition model and what the experience have been

Author Response

Reviewer 3:

Manuscript by Maria Persen et al describes the importance of a well-structured transition, from pediatric to adult care, in young adults with Prader-Willi syndrome. 

The authors state that, despite the importance of the collaboration between the pediatric and the adult team for an optimal transition, joined medical visits at the time of transition are extremely difficult given the tight schedule of the medical teams in a hospital setting. The fact of not being able to have a joined visit often results in miss information and stress and anxiety of the unprepared patients for the new situation.  To resolve the situation the authors present a new model of transition.

Concerns:

  • Please described the new model with details

Response: Thank you for this important comment. The new model has been described with details in the Result section on page 3.

  • Please give a full prospective on the transition and include the pediatric team prospective. What is the pediatric team doing at the last visit?

Response: The transition including the pediatric teams prospective has been included in the Result section on page 3.

Authors state that the main focus on the care during childhood is auxology. Please review with the pediatric team as the pediatric team main focus is mainly the prevention of co-morbidities and also treatment when needed and not auxology.

Response: Thank you for this comment. The sentence has been amended. Please see response to reviewer 1.

  • Please provide more details on important medical facts at the time of transition. Otherwise review the manuscript as role of the nurses in the transition of PW patients

Response: We thank the reviewer for this comment. The role of the nurses has been reviewed and explained in the manuscript in the Result section on page 3.

  • The new model is based on a joined visit with the pediatric and adult team? How the authors resolved the possibility of performing a joined visit with their tight schedule? Are they still having the same trouble and often end up missing that joined visit?

Response: Our apologies for the unclarity in this sentence. “specialist” was missing and is now added. Furthermore, the following has been included on page 3: “Since the transition meetings have been limited to joined meetings between the patient, relatives or caregivers, the pediatric and adult team, the meetings are now performed without problems or delays and are never missed.”

  • They have adapted a check list from the “Ready, Steady, Go” program to not miss any information. Is the pediatric team filling the check list and then the nurse from the adult team reviewing?

Response: We thank the reviewer for this comment. We have amended the sentence to: The checklist includes individual care and considerations and is filled in by the adult nurse at the last visit in the pediatric clinic. However, the checklist is easy to use by anyone.

  • Results: the results should be carefully reviewed. 1) How many patients have you transitioned using the current model?

Response: Thank you for this comment. Eight patients have been transferred according to the current model. This has been included in the text.

2) Authors state that patients are now prepared and less anxious; how have the authors measured this outcome?

Response: Thank you for the comment. This was unfortunately not measured but according to our experience, which has been explained in the text on page 3.

3) Authors state they have no dropouts with the current model; did you have any dropouts before? Please provide more information about the experience.

Response: We had no dropouts before or since we started with the current model, but the transition visits are performed at a more appropriate time. The sentence has been clarified to: “We have never had dropout from the visits in adult care, but with the current model the first visit in the adult care setting is performed more timely.

  • Please review title: “an adapted model”? is it an adapted model or an adapted check list?

Response: We thank the reviewer for this interesting comment. Our reference model is the multidisciplinary team meeting and we have an adapted model involving only a limited number of specialists.

Authors need to provide all the readers with the following information: details about the transition model and what the experience have been

Response: Thank you for the comments. The Result section has been revised and expanded accordingly.

Reviewer 4 Report

Review of Pedersen and Hoybye article on Transition to adult care in PWS

Pedersen and Hoybye draw attention to the important issue of transitioning individuals with Prader-Willi syndrome from pediatric to adult medical care. Various guidelines for transitioning the care of adolescents with chronic medical conditions to the care of adult specialists have been published. In addition to the article by Clayton et al (reference #5 in this paper), the Endocrine Society (USA) has provided specific guidelines for transition from pediatric to adult providers for patients with type 1 diabetes mellitus, growth hormone deficiency, and Turner syndrome  (https://www.endocrine.org/improving-practice/patient-resources/transitions). 

Transition for individuals with PWS is especially challenging in view of the complex nature of this syndrome which requires input from and cooperation among multiple medical specialists. Previous publications (references #3, 4, 7, 8 in this paper) describe specific details of the different protocols and effects of transition programs for PWS in various institutions.

Pedersen and Hoybye now report how they adapted the previously described guidelines for use within their “limited setting with pediatric and adult endocrinologists and nurses.” I believe that the main message in this brief narrative report is that while the concept of transition care is applicable to every institution caring for individuals with PWS, specific protocols need to be adapted and modified according to the available staff and facilities of each clinic setting.

Specific comments:

Page 2, paragraph 4: “The adult PWS team…consists of a nurse and an endocrinologist…” It would be helpful to describe the role of the nurse in some detail. Also, a dietitian should be an integral member of the management team. In fact, the same  dietitian with experience both in pediatric and adult nutrition could serve as a consistent and familiar member of both the pediatric and adult management teams

Page 3, paragraph 2: “In our hospital the endocrinologist has the main responsibility for the care of the patient with PWS…” Although there are many endocrine disorders that require treatment in adults with PWS, behavioral problems and psychiatric issues are common in almost all PWS adults. I would think that a psychiatrist and/or psychologist would need to be an integral member of the adult clinic staff.

Page 3, table 1: The table is a nice example of a checklist for the transition process, however in its current format, there is not enough space between the lines for readers, who wish to copy the form for use in their own institutions, to fill in the required information. Also, the listing of “can do, needs help with” in the first section of the table is very general.  Some examples of specific activities of daily living should be listed so that the “can do” vs “needs help with” choices can be selected for each activity.  The topics of “injections” and “medications” should include doses as well. “Meal planning” seems to be too general. Input from a dietitian providing specific caloric intake should be added. Other items missing from the table include details of sex hormone replacement, hypothyroidism, vitamin D treatment, and bone mineral density results.

Page 5, paragraph 2: “…might also be financially problems” should be “financial problems”

Author Response

Reviewer 4:

Pedersen and Hoybye draw attention to the important issue of transitioning individuals with Prader-Willi syndrome from pediatric to adult medical care. Various guidelines for transitioning the care of adolescents with chronic medical conditions to the care of adult specialists have been published. In addition to the article by Clayton et al (reference #5 in this paper), the Endocrine Society (USA) has provided specific guidelines for transition from pediatric to adult providers for patients with type 1 diabetes mellitus, growth hormone deficiency, and Turner syndrome  (https://www.endocrine.org/improving-practice/patient-resources/transitions). 

Transition for individuals with PWS is especially challenging in view of the complex nature of this syndrome which requires input from and cooperation among multiple medical specialists. Previous publications (references #3, 4, 7, 8 in this paper) describe specific details of the different protocols and effects of transition programs for PWS in various institutions.

Pedersen and Hoybye now report how they adapted the previously described guidelines for use within their “limited setting with pediatric and adult endocrinologists and nurses.” I believe that the main message in this brief narrative report is that while the concept of transition care is applicable to every institution caring for individuals with PWS, specific protocols need to be adapted and modified according to the available staff and facilities of each clinic setting.

Response: We thank the reviewer for the nice and excellent comments. They have been included in the Discussion section and in the Conclusion on page 6. Furthermore, The Endocrine Society’s guidelines have been referenced (reference #21)  

Specific comments:

Page 2, paragraph 4: “The adult PWS team…consists of a nurse and an endocrinologist…” It would be helpful to describe the role of the nurse in some detail. Also, a dietitian should be an integral member of the management team. In fact, the same dietitian with experience both in pediatric and adult nutrition could serve as a consistent and familiar member of both the pediatric and adult management teams

Response: We thank the reviewer for this comment. The role of the nurses has been reviewed and explained in the manuscript in the result section on page 3.

Page 3, paragraph 2: “In our hospital the endocrinologist has the main responsibility for the care of the patient with PWS…” Although there are many endocrine disorders that require treatment in adults with PWS, behavioral problems and psychiatric issues are common in almost all PWS adults. I would think that a psychiatrist and/or psychologist would need to be an integral member of the adult clinic staff.

Response: We thank the reviewer for this important comment. The suggestion has been included in the manuscript, also please see our response above. 

Page 3, table 1: The table is a nice example of a checklist for the transition process, however in its current format, there is not enough space between the lines for readers, who wish to copy the form for use in their own institutions, to fill in the required information.

Response: Thank you for this valuable comment. The spaces have been the lines have been expanded.

Also, the listing of “can do, needs help with” in the first section of the table is very general.  Some examples of specific activities of daily living should be listed so that the “can do” vs “needs help with” choices can be selected for each activity. 

Response: We thank the reviewer for this comment. The purpose of the checklist is to give a general overview that follow-up within important areas has been considered and reviewed. We feel the checklist becomes too busy if more text would be included. We deleted “can do” to further expand the room for comments. 

The topics of “injections” and “medications” should include doses as well.

Response: Thank you for this comment. The suggestions have been included.

“Meal planning” seems to be too general. Input from a dietitian providing specific caloric intake should be added.

Response: We completely agree. Some lines above “Meal planning” contact with a nutritionist is already included.

Other items missing from the table include details of sex hormone replacement, hypothyroidism, vitamin D treatment, and bone mineral density results.

Response: We thank the reviewer for this excellent comment. These are important medications and examinations but in our institution such information is found in the medical records and the purpose of the checklist is not to replace the medical records and the suggested items are therefore not included.   

Page 5, paragraph 2: “…might also be financially problems” should be “financial problems”

Response: Our excuses for this error, which has been corrected.

Round 2

Reviewer 3 Report

Authors have adequately answer all my comments